# Anti-Inflammatory Activity of Glycolipids and a Polyunsaturated Fatty Acid Methyl Ester Isolated from the Marine Dinoflagellate *Karenia mikimotoi*

**DOI:** 10.3390/md18030138

**Published:** 2020-02-27

**Authors:** Alain S. Leutou, Jennifer R. McCall, Robert York, Rajeshwar R. Govindapur, Andrea J. Bourdelais

**Affiliations:** Center for Marine Science, University of North Carolina at Wilmington, 5600 Marvin K. Moss Lane, Wilmington, NC 28409, USA; leutoualain@yahoo.fr (A.S.L.); mccalljr@uncw.edu (J.R.M.); yorkra@uncw.edu (R.Y.); govindapurr@uncw.edu (R.R.G.)

**Keywords:** dinoflagellate, *Karenia mikimotoi*, glycolipids, monogalactosyldiacylglycerol, monogalactosylmonoacylglycerol, polyunsaturated fatty acid methyl ester, *Staphylococcus aureus*, *Escherichia coli*, *Candida albicans*, anti-inflammatory activity

## Abstract

A new monogalactosyldiacylglycerol (MGDG), a known monogalactosylmonoacylglycerol (MGMG) and a known polyunsaturated fatty acid methyl ester (PUFAME) were isolated from the marine dinoflagellate *Karenia mikimotoi*. The planar structure of the glycolipids was elucidated using mass spectroscopy (MS) and nuclear magnetic resonance (NMR) analyses and comparisons to the known glycolipid to confirm its structure. The MGDG was characterized as 3-*O*-*β*-D-galactopyranosyl-1-*O*-3,6,9,12,15-octadecapentaenoyl-2-*O*-tetradecanoylglycerol **1**. The MGMG and PUFAME were characterized as (2*S*)-3-*O*-*β*-D-galactopyranosyl-1-*O*-3,6,9,12,15-octadecapentaenoylglycerol **2** and Methyl (3*Z*,6*Z*,9*Z*,12*Z*,15*Z*)-octadeca-3,6,9,12,15-pentaenoate **3**, respectively. The isolation of the PUFAME strongly supports the polyunsaturated fatty acid (PUFA) fragment of these glycolipids. The relative configuration of the sugar was deduced by comparisons of ^3^*J*_HH_ values and proton chemical shifts with those of known glycolipids. All isolated compounds MGDG, MGMG and PUFAME **1**-**3** were evaluated for their antimicrobial and anti-inflammatory activity. All compounds modulated macrophage responses, with compound **3** exhibiting the greatest anti-inflammatory activity.

## 1. Introduction

Microorganisms isolated from marine environments have been considered as a good source for exploration of novel natural products [1,2]. *Karenia mikimotoi* (Miyake et Kominami ex Oda) Hansen et Moestrup (former name *Gymnodinium aureolum*, *G. nagasakiense*, *G. mikimotoi*) is a harmful dinoflagellate species with a worldwide distribution that forms massive blooms in coastal waters in the temperate regions of Europe [3,4,5,6], East Asia [7] and the east coast of United States of America [8]. *K. mikimoitoi* has been associated with fish kills around the globe. The toxic components produced by *K. mikimotoi* that have been associated with fish kills are hemolysins. Hemolysins, are typically uncharacterized substances that can lyse red blood cells (RBCs) by altering membrane permeability [9]. Other components of *K. mikimoitoi* blooms that have a negative effect on co-occurring species include: lipids, sterols and/or PUFAs [8,10]. Additionally, digalactosylmonoacylglycerols and PUFAs isolated from *K. mikimotoi* have been shown to be hemolytic and ichthyotoxic [11].

Monogalactosyldiacylglycerols (MGDGs), monogalactosylmonoacylglycerols (MGMGs) and digalactosyldiacylglycerols (DGDGs) have attracted much attention in recent years because of their biological activities, such as anti-tumor-promoting [12,13], oxygen scavenging [14], anti-viral [15,16], anti-inflammatory [17] and anti-hyperlipidemic [18] properties. Nagatsu et *al*. demonstrated that MGDGs were more potent than MGMGs and that MGDGs with a myristoyl group at sn-1 position were more potent than those at sn-2 position, although MGDGs with highly unsaturated acyl groups, which have a myristoyl group at sn-2 position, showed strong effects [19]. The length of the acyl chain was found to be more important for the activity of MGMGs rather than the position of the ester function. For example, acyl chain length of six carbon atoms resulted in the maximum anti-tumor activity in synthetic MGMGs with longer and shorter chain lengths showing less activity regardless of the position of the myristol group [20]. 

Glycoglycerolipids are usually available from natural sources in only limited quantities and it is often difficult to separate them from mixtures [21]. In nature, glycoglycerolipids are produced by dinoflagellates by their plastids. The evolutionary occurrence of plastids in some dinoflagellates is believed to result from the secondary endosymbiosis of a red algal plastid-like ancestor. Other photosynthetic dinoflagellates gained plastids through a tertiary endosymbiosis with another algal partners such as haptophytes, as in the case of *Karenia* and *Karlodinium* [22]. The glycerolipids produced by plastids are used by photosynthetic dinoflagellate in their chloroplast membranes. The cholorplast membranes are composed principally of two galactolipids, mono-and digalactosyldiacylglycerol (MGDG and DGDG, respectively). It has been found that the peridinin-containing dinoflagellates can be divided into two clusters depending on whether the major forms of MGDG and DGDG are C18/C18 (sn-1/sn-2 regiochemistry) or C20/C18 containing octadecatetraenoic acid 18:5n-3 or eicosapentaenoic acid 20:5n-3 [23]. Fatty acids such as 18:5n-3 and 14:0 are found to be key components of the MGDGs and MGMGs in the dinoflagellate *Karenia brevis*. Another dinoflagellate, *Lepidodinium chlorophorum*, was found to possess glycoglycerolipids with fatty acid (FA) containing side chains of 18:5/16:4, 18:5/18:5 for a MGDG and 20:5/16:4 for a DGDG [24].

During the investigation of the *K. mikimotoi* strain CC21, two glycoglycerolipids an MGDG and an MGDG with the FA compositions of 18:5n-3 and 18:5n-3/14:0 and one PUFAME with the FA composition of 18:5n-3 were found, demonstrating similar lipid structures as seen in the dinoflagellates *K. brevis* and *L. chlorophorum* [24]. Investigation of the lipids found in *K. mikimotoi* resulted in the identification of a new galactolipid, 3-*O*-*β*-D-galactopyranosyl-1-*O*-3,6,9,12,15-octadecapentaenoyl-2-*O*-tetradecanoylglycerol **1**; a known galactolipid (2*S*)-3-*O*-*β*-D-galactopyranosyl-1-*O*-3,6,9,12,15-octadecapentaenoylglycerol **2** [25] and a known PUFAME, methyl (3*Z*,6*Z*,9*Z*,12*Z*,15*Z*)-octadeca-3,6,9,12,15-pentaenoate **3** (Figure 1) [25]. 

## 2. Results and Discussion

Monogalactosyldiacylglycerol **1** was obtained as an optically active amorphous white solid, (${\left[\alpha \right]}_{D}^{25}$ −16.7, *c* 0.30, MeOH). The molecular formula of **1** was established as C_41_H_68_O_10_ on the basis of high resolution electrospray ionization time of flight (HRESITOF) mass spectroscopy (*m/z* 743.4727 [M + Na]^+^ (calcd for C_41_H_68_O_10_Na, 743.4710)) (Appendix A). The IR spectrum of **1** revealed absorptions indicative of hydroxyl (3361 cm^−1^), ester (1736 cm^−1^) and glycosidic moieties (1085 cm^−1^). **1** was submitted to electrospray ionization mass spectrometry (ESI-MS). The positive ion mode gave the ions at *m/z* 721.9 [M + H]^+^, 703.9 [M + H-H_2_O]^+^ (Appendix A). Analyses of the correlation spectroscopy (COSY) together with heteronuclear single quantum correlation (HSQC) and heteronuclear multiple-bond coherence (HMBC) spectra led to the identification of a terminal *β*-galactopyranose unit with ^13^C NMR signals at *δ*_C_ 105.4 (C-1‴), 72.5 (C-2‴), 75.0 (C-3‴), 70.3 (C-4‴), 76.9 (C-5‴) and 62.5 (C-6‴), and a glycerol moiety with carbon signals at *δ*_C_ 64.5 (C-1), 71.8 (C-2) and 68.8 (C-3). In its ^1^H NMR spectrum, signals were observed corresponding to two methyls at *δ*_H_ 0.90 (3H, t, *J* = 7.3 Hz) and 0.98 (3H, t, *J* = 7.6 Hz), numerous oxymethylenes and oxymethines between *δ*_H_ 3.36 and 4.44, one oxymethine at *δ*_H_ 5.27 (m, H-2), methylenic hydrogens between *δ*_H_ 1.20 and 3.15 and olefinic hydrogens between *δ*_H_ 5.35 and 5.56. The ^13^C and ^1^H NMR (Table 1) signals of **1** were similar to those of (2*S*)-3-*O*-*β*-D-galactopyranosyl-1-*O*-3,6,9,12,15-octadecapentaenoylglycerol [25], with the exception of signals at *δ*_C_ 174.6, 35.2, 26.1, 30.6, 30.2–30.8, 33.1, 23.8, 14.5 and *δ*_H_ 2.33 (t, *J* = 7.4 Hz, H-2″), 1.61 (m, H-3″), 1.33 (m, H-4″-H13″) and 0.90 (t, *J* = 7.3 Hz, H-14″), suggesting the presence of 2-*O*-tetradecanoyl moiety. The anomeric H-1‴ proton of **1** at *δ*_H_ 4.22 (d, *J* = 7.5 Hz) showed HMBC cross-peaks with C-3 (*δ*_C_ 68.8) of the glycerol moiety and C-2‴ (*δ*_C_ 72.5). The protons H-1a,b at *δ*_H_ [4.25 (dd, *J* = 12.1, 6.3 Hz) and 4.43 (dd, *J* = 12.1, 3.0 Hz)] of the glycerol was correlated to the carbonyl C-1’ (*δ*_C_ 173.1) of the fatty acid side chain. The 2-*O*-tetradecanoyl moiety was attached at C-2 by the proton chemical shift of C-2, *δ*_H_ (m, 5.27), which appears in the down field shift. The observation of the fragment ion peak at *m/z* 331 corresponded to the loss of tetradecanoic acid and the sugar moiety, which clearly supports the PUFAs description 3,6,9,12,15-octadecapentaenoic acid; this PUFA moiety was also support by observation of an ion peak at *m/z* 257. The tetradecanoic acid was further support by fragment ion peak at *m/z* 285, which corresponded to the loss of 3,6,9,12,15-octadecapentaenoic acid and a sugar moiety (Figure 2) and Appendix A). The overall analyses determined the structure of **1** as shown in (Figure 3). The anomeric *β*-configuration of the glycosidic bond was determined on the basis of the anomeric signal at *δ*_C_ 105.4 (C-1‴) [26]. Bis-allylic carbon signals of *Z* and *E*-isomers are observed at *δ*_C_ ca. 27 and ca. 32, respectively [25,27,28]; the 26.6 ppm shift suggests that all double bonds have a *cis* geometry (*Z*). Because of the small amount of monogalactosyldiacylglycerol **1** produced by *K. mikimotoi*, we were able to confirm the exact nature of the polyunsaturated fatty acids, by isolation of methyl (3*Z*,6*Z*,9*Z*,12*Z*,15*Z*)-octadeca-3,6,9,12,15-pentaenoate in this strain. **1** was subjected to fragmentation mass spectroscopy; the observations of the fragment ion peak at *m/z* 229 corresponded to a tetradecanoic fatty acid. The fragment ion peak at *m/z* 559 corresponded to the loss of a sugar moiety. 

A monogalactosylmonoacylglycerol (MGMG) **2** was obtained as an optically active amorphous white solid, (${\left[\alpha \right]}_{D}^{25}$ −62.9, *c* 0.06, MeOH). The infrared (IR) spectrum of **2** revealed absorptions indicative of hydroxyl (3366 cm^−1^), ester (1726 cm^−1^) and glycosidic moieties (1084 cm^−1^). The molecular formula of **2** was established as C_27_H_42_O_9_ on the basis of (HRESITOF) mass spectroscopy (*m/z* 533.2727 [M + Na]^+^ (calcd for C_27_H_42_O_9_Na, 533.2727)) (Appendix A). Comparison of NMR spectroscopic data of **2** to those of previously reported spectra showed that the planar structure of **2** was identified as the know MGMG (2*S*)-3-*O*-β-d-galactopyranosyl-1-*O*-3,6,9,12,15-octadecapentaenoylglycerol [25]. 

Methyl (3*Z*,6*Z*,9*Z*,12*Z*,15*Z*)-octadeca-3,6,9,12,15-pentaenoate **3** was obtained as a white solid. Its molecular formula C_19_H_28_O_2_ was determined based on ^1^H and ^13^C spectroscopic data (NMR) and HRESITOF mass spectroscopy data (observed [M + H]^+^ ion at *m/z* 289.2169, calculated [M + H]^+^ ion at 289.2168) (Appendix A). Comparison of NMR spectroscopic data of polyunsaturated fatty acid methyl ester **3** to those of the reported one permitted that the planar structure of **3** was identified as methyl (3*Z*,6*Z*,9*Z*,12*Z*,15*Z*)-octadeca-3,6,9,12,15-pentaenoate [25,29,30,31]. The isolation of compound **3** is a good support of the structure of a new monogalactosyldiacylglycerol (MGDG) **1**.

All three compounds were assessed for antimicrobial activity on the following potential human pathogens: gram positive bacteria *Staphylococcus aureus*, gram negative bacteria *Escherichia coli* and fungi *Candida albicans.* None of the compounds exhibited the ability to kill or inhibit growth of either bacterial or fungal species (results not shown). 

All three compounds were assessed for immunomodulatory potential on the RAW 264.7 macrophage cell line exposed to the inflammatory stimulus lipopolysaccharide (LPS). Cells were exposed to LPS for 72 h prior to the assay, with treatment with the compounds 24 hour prior to testing (48 h after LPS). As can be seen in Figure 4, the compounds had varying effects on cell parameters. Compounds **2** and **3** both alter cell size (FSC) and complexity (SSC), causing smaller but more complex cells. Cell surface markers of macrophage activation indicate that compound **1** increased TLR 4 expression, but decreased CD80 expression (as did compound **3**). CD80 is a marker of M1, or classical, activation in macrophages. Classically activated M1 macrophages, as seen with an initial response to infection, produce pro-inflammatory cytokines to coordinate other immune cells in a battle against an infection or injury [32,33,34]. While none of the compounds had an effect on CD206, all three significantly reduced expression of CD124. Both CD206 and CD124 are cell surface markers of the M2 activation state in macrophages. M2 macrophages are considered the alternatively activated and are associated with wound and tissue healing [32,33,34]. As CD124 is a receptor for the signaling cytokine IL-4, this result indicates that these compounds are reducing macrophage ability to respond to this molecule. In particular, compound **3** appears to have the most aggregate effects, decreasing M1 activation (CD 80) and M2 activation (CD124), in addition to decreasing size and increasing complexity of RAW 264.7 cells.

To assess the functional response to decreases in activation states seen by these compounds on macrophages, media was collected from RAW 264.7 cells treated with LPS for 72 total h and the test compounds for the 24 h after the LPS treatment LPS treatment. As seen in Figure 5, expression of the pro-inflammatory cytokine TNFα was significantly decreased in response to treatment with compound **3** alone. This functional response is in agreement with the flow cytometry data (Figure 4), where compound **3** had the most effects on macrophage expression of activation markers and size/complexity. Compound 3 resulted in an approximately 22% decrease in TNFα secretion, which is similar to what has been found from brevenal, another natural product, derived from *Karenia brevis* [35].

Interestingly, none of the compounds altered expression of the anti-inflammatory IL-10. IL-10 is typically associated with an M2 phenotype, as it suppresses immune responses and prevents tissue damage due to inflammation [36]. This data indicates that compound **3** may have potential as an anti-inflammatory therapy if it can reduce activation and inflammatory cytokine secretion while maintaining IL-10 secretion and thus its protective effects, though further studies would need to be conducted to confirm this hypothesis.

One may expect, with increased expression of TLR4 (Figure 4C), that compound **1** would allow cells to be more responsive to LPS treatment. However, the increase in TNFα secretion (Figure 5A) was not statistically significant, and it could be that the concomitant decrease in CD80 expression (Figure 4D) results in a net neutral activation state for these macrophages.

## 3. Experimental Section

### 3.1. Strain and Cultivation

*Karenia mikimotoi*, strain ARC 163, was isolated from the marine waters off Corpus Christi, TX (27.84567,−97.429848). A voucher specimen was deposited by L. Campbell, (2006) and is maintained at the Algal Resources Collection (ARC), University North Carolina Wilmington. The *Karenia* species was identified on the basis of morphology by ARC. The strain was grown in two 10 L batch static cultures with LH media, (NaNO_3_ 75 g/L of pyrogen free DIW, NaH_2_PO_4_.H_2_O 5 g/L, Na_2_SeO_3_ 0.0045 g/L, PII Metal Mix, F/2 Vitamin Solution) in 33SU at 22 °C, light intensity of 70 µE s^−1^ m^−2^ and photoperiod of 8:16 for 25–30 days. The finial density was approximately 2.0 × 10^7^ cells/L.

### 3.2. Extraction and Isolation

After approximately 30 days under static growth, the whole culture was extracted with ethyl acetate (EtOAc), with EtOAc-Water (1:1 v:v) and evaporated under reduced pressure to yield the crude organic extract (0.4 g). 

The ethyl acetate extract was then partitioned between 90% methanol in water and petroleum ether (1:1 v:v) to separate non-polar lipids from polar lipids. The methanol-soluble layer was removed and concentrated under vacuum to give 0.2 g dry weight. The methanol-water layer was then subjected to reversed-phase HPLC (Agilent, Zorbax C_8_, 5 µm, 250 × 9.4 mm, 3.4 mL/min). Elution was performed with Water-MeOH gradient elution (stepwise, 50–100% MeOH) to yield eight fractions. The H_2_O/MeOH (2:98) fraction contained a mixture of metabolites, which was further fractionated by reversed-phase HPLC (Phenomenex Luna Phenyl-Hexyl, 250 × 4.6 mm, 1.4 mL/min, 5 *μ*m, UV detection at 215 nm) using an isocratic solvent system of 89% MeOH 11% H_2_O to separate three compounds: monogalactosyldiacylglycerol (**1**, t_R_ = 20 min, 0.7 mg), monogalactosylmonoacylglycerol (**2**, t_R_ = 17 min, 0.3 mg) and PUFAME (**3**, t_R_ = 6 min, 0.4 mg).

### 3.3. Structure Elucidation

The optical rotation was measured using an Autopol III (Rudolph Research) polarimeter with a 1 cm cell. IR spectra were recorded on a Bruker Fourier transform (FT)-IR model IFS-88 spectrometer. NMR spectra were obtained with Bruker 500 MHz spectrometer in *d*4-methanol, using the signals of the residual solvent protons and the solvent carbons as internal references (*δ*_H_ 3.30 and *δ*_C_ 49.0 ppm for CD_3_OD). HRESITOF-mass spectra were measured on a Waters Xevo G2-XS QTOF mass spectrometer. The instrument was scanned from 200 to 1500 *m/z* with the spectra shown consisting of the sum of 20 scans. Samples for MS analysis were dissolved in either 50:50 water/acetonitrile or 49:49:2 water/acetonitrile/acetic acid before being directly injected at a flow rate of 10 *µ*L/min. Low-resolution electrospray ionization mass spectrometry (LR-ESI-MS) data were measured using a Sciex QTRAP 4000 LC/MS system. Samples were dissolved in 98% acetonitrile 2% water and 0.1% formic acid and directly infused into the mass spectrometer using a Harvard syringe pump with a flow rate of 10 μL/min. The mass spectrometer parameters were: declustering potential 80V, Ion spray voltage 5000 V, Temperature 350 °C, Gas Source 1 35 psi, gas source 2 35 psi, interface heater on). Fragments of the target compounds were produced by ramping the collision energy from 5–130 V. 

HPLC was conducted on a SHIMADZU LC-10AD Pump coupled with a SHIMADZU SPD-10A UV/vis detector. The UV wavelength used for separation was 215 nm. The flow rate of the mobile phase was 3.4 mL/min for 9.4 mm column (Agilent, Zorbax C_8_, 250 × 9.4 mm, 5 µm) and 1.4 mL/min for 4.6 mm column (Phenomenex Luna Phenyl-Hexyl, 250 × 4.6 mm, 5 *μ*m).

#### 3.3.1. Monogalactosyldiacylglycerol **1**

Optically active amorphous white solid; (${\left[\alpha \right]}_{D}^{25}$ −16.7, *c* 0.30, MeOH); IR (KBr) *_ν_*_max_ 3361, 2923, 2853, 1736, 1277, 1163, 1085 cm^−1^; ^1^H and ^13^C NMR data (Table 1); HRESITOF *m/z* 743.4727 [M + Na]^+^ (calcd for C_41_H_68_O_10_Na, 743.4710).

#### 3.3.2. Monogalactosylmonoacylglycerol **2**

Optically active amorphous white solid; (${\left[\alpha \right]}_{D}^{25}$ −62.9, *c* 0.06, MeOH); IR (KBr) *_ν_*_max_ 3366, 2920, 2847, 1726, 1476, 1084 cm^−1^; HRMSESITOF *m/z* 533.2727 [M + Na]^+^ (calcd for C_27_H_42_O_9_Na, 533.2727); ^1^ H NMR (500 MHz, CD_3_OD): *δ*_H_ 5.55 (2H, q, *J* = 4.9 Hz, H-3’, H-4’), 5.39 (1H, m, H-16’), 5.36 (7H, m, H-6’, H-7’, H-9’, H-10’, H-12’, H-13’, H-15’), 5.28 (1H, m, H-2), 4.43 (1H, dd, *J* = 12.1, 3.0 Hz, H-1b), 4.27 (1H, dd, *J* = 12.1, 6.3 Hz, H-1a), 4.23 (1H, d, *J* = 7.5, H-1‴), 3.99 (1H, dd, *J* = 10.9, 5.4, H-3b), 3.82 (1H, brd, *J* = 3.2, H-4‴), 3.77 (1H, m, H-3a), 3.75 (2H, m, H-6‴), 3.51 (1H, m, H-2‴), 3.51 (1H, m, H-5‴), 3.44 (1H, dd, *J* = 9.7, 3.2, H-3‴), 3.15 (2H, dd, *J* = 7.9, 5.9 Hz, H-2’), 2.84 (8H, m, H-5’, H-8’, H-11’, H-14’), 0.97 (3H, *J* = 7.6, H-18’); ^13^C NMR (125 MHz, CD_3_OD): *δ*_C_ 172.0 (C-1’), 132.9 (C-16’), 132.6 (C-4’), 129.7 (X3), 129.3, 128.3, 128.2 (X2) (C-6’, C-7’, C-9’, C-15’, C-13’, C-12’, C-10’), 122.5 (C-3’), 105.3 (C-1‴), 76.9 (C-5‴), 74.9 (C-3‴), 72.1 (C-2‴), 70.4 (C-4‴), 70.4 (C-2), 67.0 (C-3), 62.6 (C-6‴), 62.6 (C-1), 32.3 (C-2’), 26.7, 26.6 (X2), 26.5 (C-5’, C-8’, C-11’, C-14’), 21.6 (C-17’), 14.7 (C-18’) (Appendix A), in agreement with data reported in the literature [25].

#### 3.3.3. Methyl (3*Z*,6*Z*,9*Z*,12*Z*,15*Z*)-octadeca-3,6,9,12,15-pentaenoate **3**

White solid; ^1^H NMR (500 MHz, CD_3_OD): 5.50-5.59 (2H overlapping, m, alkene-H); 5.24-5.43 (8H overlapping, m, alkene-H), 3.13 (2H, d, *J* = 5.3 Hz, H-2); 3.67 (3H, s, -OCH_3_); 2.78-2.84 (four overlapping t, 8H, m, H-5, H-8, H-11, H-14), 2.08 (2H, quintet, *J* = 7.6, H-17); 0.97 (3H, t, *J* = 7.6, H-18); ^13^C NMR (125 MHz, CD_3_OD): *δ*_C_ 172.0 (C-1), 132.8 (C-16), 132.5 (C-4), 129.6, 129.5, 129.3, 129.0, 128.9, 128.4 (C-6, C-7, C-9, C-10, C-12, C-13), 128.2 (C-15), 122.4 (C-3), 52.3 (COOCH_3_), 33.4 (C-2), 26.4, 26.5, 26.5, 26.6 (C-5, C-8, C-11, C-14), 21.4 (C-17), 14.6 (C-18) (Appendix A), in agreement with data reported in the literature [25,29,30,31]. HRESITOF *m/z* 289.2169 [M+H]^+^ (calcd for C_19_H_29_O_2_, 289.2168), LR-ESI-MS *m/z*: 288 [M]^+^, 259 [M-Et]^+^.

### 3.4. Antimicrobial Bioassay Testing

Compounds were tested for antimicrobial activity using a Minimum Inhibitory Concentration (MIC) assay with the gram-positive bacteria *Staphylococcus aureus*, the gram-negative bacteria *Escherichia coli*, and the fungi *Candida albicans* (Carolina Biological, Greensboro, NC, USA). Bacteria were inoculated in nutrient broth for 24 h at 37 °C, and fungi were inoculated in Sabourand broth for 24 h at 27 °C.

Test compounds and positive controls (10–50 µg/mL doxycycline for bacteria and 50 µg/mL amphotericin B for fungi) were dissolved in methanol, and 100 μL were added to duplicate wells of a 96-well assay plate. Compounds and controls were serially diluted 1:2 in broth. Vehicle control (methanol only) and negative control (no compound) was added to control wells. Following compound/control dilutions, 10 μL of bacteria or fungi were added to each well for a final concentration of 5 × 10^4^ CFUs per well. Plates were incubated for 24 h at the corresponding temperature (37 °C for bacteria, 27 °C for fungi). Plates were read for OD at 600 nm to assay for bacteriostatic or fungistatic activity. To assess for bactericidal or fungicidal activity, wells were assayed using an XTT Cell Proliferation Assay Kit (ATCC, Manassas, VA, USA), according to the manufacturer’s instructions.

### 3.5. Flow Cytometry and ELISA Experiments

RAW 264.7 macrophages were seeded in 12-well plates and incubated at 37 °C until cells adhered and grew to confluence before being treated with LPS (50 ng/mL). Cells were incubated for 48 h prior to treatment with vehicle control (EtOH) or one of the three compounds for another 24 h (72 total h LPS). Cells were then harvested, centrifuged, and media removed for ELISA experiments. Cells were blocked with anti-goat serum to block Fc receptors for 15 min on ice, then stained for 1 h with TLR4, CD80, CD206 and CD124 antibody solution. Unbound antibody was removed and cells resuspended in cold phosphate buffered saline prior to analysis on a BD FACS Celesta flow cytometer (BD Biosciences; San Jose, CA, USA). 

Expression of Toll-like receptor 4 (TLR4 or CD284/MD-2) was measured using BV650-conjugated rat anti-mouse antibodies directed against CD284/MD-2 (BD Biosciences; San Jose, CA, USA). Expression of mannose receptors (CD206) was measured using Alexa Fluor 488-conjugated rabbit anti-mouse antibodies directed against CD206 (Abcam; Cambridge, MA, USA). Expression of CD80 receptors was measured using phycoerythrin (PE)/CF594-conjugated hamster anti-mouse antibodies directed against CD80 (BD Biosciences; San Jose, CA, USA). Expression of IL4Rα/CD124 receptors was measured using Alexa Fluor 647-conjugated rat anti-mouse antibodies directed against CD124 (BD Pharmingen; San Diego, CA, USA).

Cell media was assayed for TNFα or IL-10 expression using commercially available ELISA kits (R&D systems; Minneapolis, MN, USA) according to the manufacturer’s directions.

## 4. Conclusions

In our aim for isolation of a new natural product, the glyceroglycolipids composition of marine dinoflagellate *K. mikimotoi* was investigated. *K. mikimotoi* is a promising candidate strain for sustainable production of high value co-products including long chain polyunsaturated fatty acids and omega-3 fatty acids. A new monogalactosyldiacylglycerol **1** was isolated and characterized. To the best of our knowledge this is the first time monogalactosylmonoacylglycerol **2** and a polyunsaturated fatty acid methyl ester **3** were isolated from *K. mikimotoi.* Compound **3** may be an artifact of the isolation procedure, and may be formed by methanolysis of biological ester. Because of a limited amount of compound **1**, we were not able to perform an acid hydrolysis; fortunately, the isolation of compound **3** further supported the fatty acid side chain of compounds **1** and **2**. The biological activities of **1**, **2** and **3** were investigated on three human pathogenic microorganisms. None of the compounds exhibited the ability to kill or inhibit growth of bacterial species or fungal species, but they did exhibit the ability to modulate macrophage activation and functionality, indicating a potential anti-inflammatory role for this family, without the negative side effects) observed with steroids (Cushing’s syndrome and bone loss) and non-steroidal anti-inflammatory drugs (gastrointestinal problems and liver toxicity) when used to treat inflammation. If the active PUFAME isolated from *K. mikimotoi* proves to be an interesting anti-inflammatory drug target, the yield of the PUFAME from the dinoflagellate could be increased through two methods. First, the growth conditions could be manipulated for higher lipid yields. Both Leblond et al. and Okuyama et *al*. have shown that cold adapted dinoflagellates have higher abundances of 18:5n-3 and 22:6n-3 PUFAMEs than do warm adapted dinoflagellates [37,38]. Second, *K. mikimotoi* produces an abundance of MGDGs and MGMGs with the fatty acid side chain of 18:5. The glyceroglycolipids could be isolated and then undergo methanolic hydrolysis to liberate the FA as the target PUFAME, which would also increase the yield.

## Figures and Tables

**Figure 1 marinedrugs-18-00138-f001:**
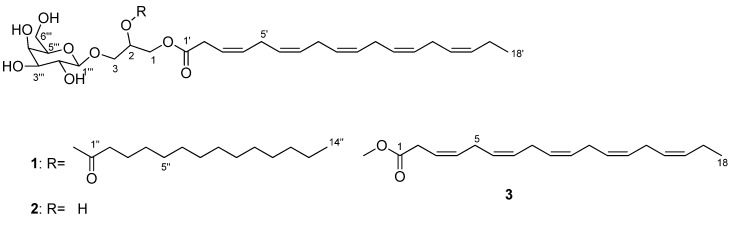
Chemical structures of **1**–**3**.

**Figure 2 marinedrugs-18-00138-f002:**
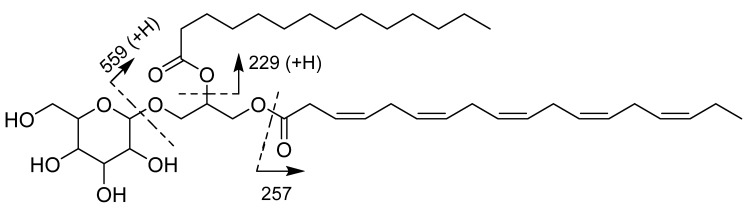
Positive (ESI-MS) of the monogalactosyldiacylglycerol **1** from *Karenia mikimotoi.*

**Figure 3 marinedrugs-18-00138-f003:**
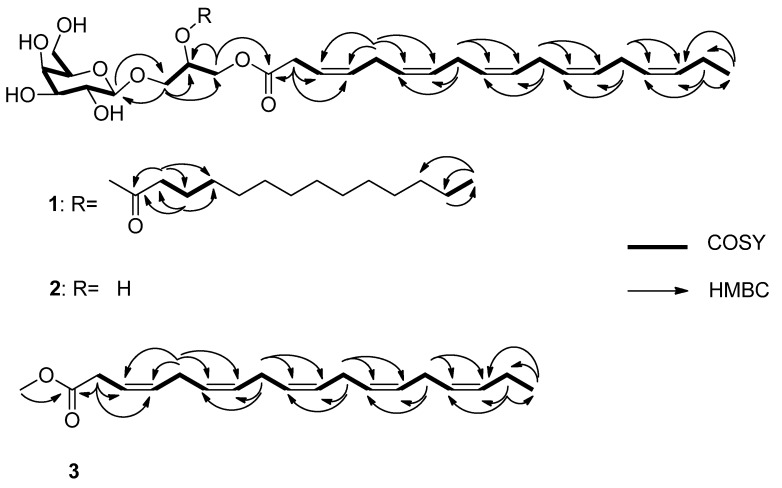
COSY and key HMBC correlations of the lipids isolated from *K. Mikimotoi* MGDG **1**, MGMG **2** and PUFAME **3**.

**Figure 4 marinedrugs-18-00138-f004:**
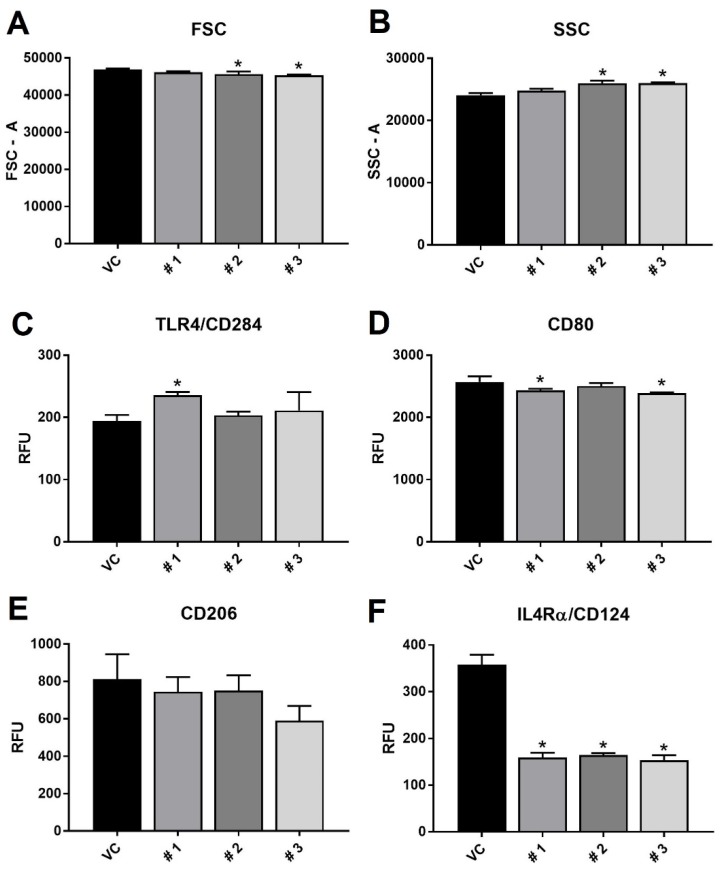
Flow cytometry results of forward scatter (FSC—panel (**A**)), side scatter (SSC—panel (**B**)), and marker/receptor expression of TLR4 (panel (**C**)), CD80 (panel (**D**)), CD206 (panel (**E**)) and IL4Rα (panel (**F**)). RAW 264.7 macrophages were treated with LPS, then exposed to vehicle control (VC) or one of the three compounds. Results are presented as average relative fluorescent units ± standard deviation, and * indicates statistical significance from control of *p* < 0.05 (n = 3).

**Figure 5 marinedrugs-18-00138-f005:**
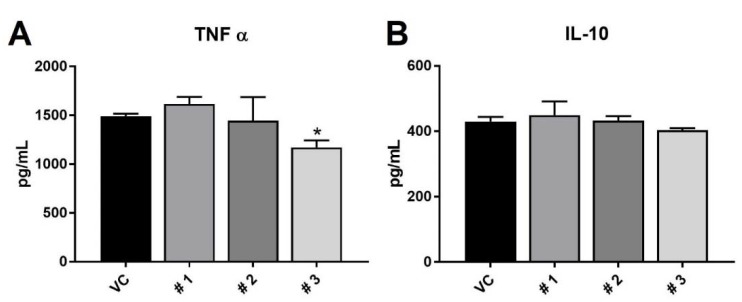
Results of cytokine secretion from LPS-stimulated RAW 264.7 macrophages after treatment with the three compounds or vehicle control (VC). Panel (**A**) shows TNFα secretion and panel (**B**) shows IL-10 secretion. Results are presented as average expression in pg/mL ± standard deviation, and * indicates statistical significance from control of *p* < 0.05 (n = 3), as determined by ANOVA.

**Table 1 marinedrugs-18-00138-t001:** NMR spectroscopic data for monogalactosyldiacylglycerol **1** in CD_3_OD, *δ* in ppm ^a^.

1				
No.	*δ*_C_, mult. ^b^	*δ*_H_ (*J* in Hz)	COSY	HMBC
1, Gly	64.5, CH_2_	4.25, dd (12.1, 6.3)	3, 2	2, 1’
		4.43, dd (12.1, 3.0)		
2, Gly	71.8, CH	5.27, m	1, 3	
3, Gly	68.8, CH_2_	3.74, m	2, 1	2, 1, 1‴
		4.00, dd (10.7, 4.0)		
1’	173.1, C			
2’	33.2, CH_2_	3.14, d (4.9)	3’	1’, 3’, 4’
3’	132.6, CH	5.55, t (4.9)	2’	
4’	122.4, CH	5.55, t (4.9)	5’	
6’-7’	128.3/129.7, CH	5.36, m	5’/8’	
9’-10’	128.3/129.7, CH	5.36, m	8’/11’	
12’-13’	128.3/129.7, CH	5.36, m	11’/14’	
15’	129.3, CH	5.36, m	14’	
16’	132.9, CH	5.36, m	17’	
17’	21.6, CH_2_	2.09, quint (7.6)	16’, 18’	15’, 16’, 18’
18’	14.7, CH_3_	0.98, t (7.6)	17’	16’, 17’
5’	26.8^c^, CH_2_	2.84, m	4’, 6’	3’, 4’, 6’, 7’
8’	26.7^c^, CH_2_	2.84, m	7’, 9’	6’, 7’, 9’, 10’
11’	26.6^c^, CH_2_	2.84, m	10’, 12’	9’, 10’, 12’, 13’
14’	26.6^c^, CH_2_	2.84, m	13’, 15’	12’, 13’, 15’, 16’
1″	174.6, C			
2″	35.2, CH_2_	2.33, t (7.4)	3″	1″, 3″, 4″
3″	26.1, CH_2_	1.61, m	2″, 4″	1″, 2″, 4″
4″	30.6, CH_2_	1.33, m	3″	
5″-11″	30.2-30.8, CH_2_	1.33, m		
12″	33.1, CH_2_	1.33, m		
13″	23.8, CH_2_	1.33, m	14″	14″
14″	14.5, CH_3_	0.90, t (7.3)	13″	12″, 13″
1‴	105.4, CH	4.22, d (7.5)	2‴	3, 2‴
2‴	72.5, CH	3.52, m	1‴	3‴
3‴	75.0, CH	3.45, dd (9.7, 3.2)	4‴	2‴
4‴	70.3, CH	3.82, brd (3.2)	3‴, 5‴	2‴
5‴	76.9, CH	3.51, m	4‴, 6‴	1‴, 4‴, 6‴
6‴	62.5, CH_2_	3.75, m	5‴	4‴, 5‴

^a^ 500 MHz for ^1^H NMR and 125 MHz for ^13^C NMR. ^b^ Numbers of attached protons were determined by analysis of two-dimensional (2D) and Distortionless Enhancement by Polarization Transfer (DEPT)-135 spectroscopic data. ^c^ Interchangeable carbons.

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
