# Peer review of "Anti-Inflammatory Activity of Glycolipids and a Polyunsaturated Fatty Acid Methyl Ester Isolated from the Marine Dinoflagellate Karenia mikimotoi"

_marinedrugs, 2020, doi:10.3390/md18030138_

Round 1

Reviewer 1 Report

This work is interesting and well presented. Minor comments: 

Check lines 107-109 (compound 2 is a MGMG) 

In line 196 it could be useful if you provide the light intensity

Some acronyms should be explained (e.g. COSY, HSQC, HMBC) 

Author Response

Response to Reviewers

Reviewer 1

R1: This work is interesting and well presented. Minor comments: 

R1: Check lines 107-109 (compound 2 is a MGMG) 

Author’s response: these lines were reworded for clarity.

R1: In line 196 it could be useful if you provide the light intensity

***Author’s response: added to the text: light intensity which the K. mikimotoi was grown under was ~70 mE s-1 m-2

R1: Some acronyms should be explained (e.g. COSY, HSQC, HMBC)

Author’s response: the acronyms were explained

Reviewer 2 Report

The authors describe isolation, structure determination and anti-inflammatory activity of a monogalactosyldiacylglycerol (MGDG), a monogalactosylmonoacylglycerol (MGMG) and a polyunsaturated fatty acid methyl ester from the marine dinoflagellate Karenia mikimotoi. Compound 1 is a new MGDG, however, there are many reports on isolation of several types of MGDGs, and anti-inflammatory property of MGDG has been already reported (ref 17). Authors should describe the originality and novelty of this work.

Did author determine the absolute configuration at C-2? If not, the structure of compound 1 in Figure 1 must be changed.

There were many errors in font style (bold or italic) throughout the manuscript. For example, numbers of compounds should be bold, and “O” and “Z” in names of compounds should be italic in line55-57.

Authors should describe on the statistical analysis for Figure 4 and Figure 5. Does (n=3) mean three independent experiments or triplicate?

I could not understand whether anti-inflammatory effects of isolated compounds were high or not. Positive control compound would be helpful for readers to understand the potency of isolated compounds 1-3 for Figure 4 and Figure 5. In addition, description on the difference of anti-inflammatory effects between known compound(s) and compounds 1-3 was needed to understand the originality and novelty of this work.

Author Response

Reviewer 2

R2: The authors describe isolation, structure determination and anti-inflammatory activity of a monogalactosyldiacylglycerol (MGDG), a monogalactosylmonoacylglycerol (MGMG) and a polyunsaturated fatty acid methyl ester from the marine dinoflagellate Karenia mikimotoi. Compound 1 is a new MGDG, however, there are many reports on isolation of several types of MGDGs, and anti-inflammatory property of MGDG has been already reported (ref 17). Authors should describe the originality and novelty of this work.

Authors response: Experiments in reference 17 referred to the anti-edema effects of MGDG in a mouse model.  MGDG treatment in ref 17 reduced swelling/fluid accumulation in vivo.  Reduced swelling is just one type of anti-inflammatory action.  These observed effects are quite broad and non-specific, and they could be attributable to any number of mechanisms of action.  Our experiments are vastly different, examining cellular mechanisms of action on key immune sentinel cells (macrophages), which coordinate inflammatory responses through, in part, inflammatory and anti-inflammatory cytokine release.  These results have not been previously reported for MGDG.

 R2: Did author determine the absolute configuration at C-2? If not, the structure of compound 1 in Figure 1 must be changed.

Authors Response: the absolute configuration was not determined due to lack of sufficient compound. The compound name has been changed to reflect this.

R2: There were many errors in font style (bold or italic) throughout the manuscript. For example, numbers of compounds should be bold, and “O” and “Z” in names of compounds should be italic in line55-57.

Authors Response:  the numbers of the compounds have been change to bold font and the O and Z have been changed to italics.

R2: Authors should describe on the statistical analysis for Figure 4 and Figure 5. Does (n=3) mean three independent experiments or triplicate?

Author’s Response:

  • The statistical analysis was ANOVA (added to figure legends)
  • N=3 represents three independent wells of cells treated with compounds or vehicle control, then analyzed separately. These three well experiments were averaged for graphical presentation and statistical analysis.

R2: I could not understand whether anti-inflammatory effects of isolated compounds were high or not. Positive control compound would be helpful for readers to understand the potency of isolated compounds 1-3 for Figure 4 and Figure 5. In addition, description on the difference of anti-inflammatory effects between known compound(s) and compounds 1-3 was needed to understand the originality and novelty of this work.

Author’s Response:

  • We appreciate the suggestion of comparing these compounds to an existing anti-inflammatory substance. Unfortunately, there is no good direct comparison of a known anti-inflammatory drug (or positive control) to these types of compounds, because the structure and mechanisms of action of corticosteroids or NSAIDs are very different.  Corticosteriods can reduce cytokine secretion to zero following LPS stimulation, but they also have a myriad of other negative side-effects, including increased susceptibility to infection due to massive immunospressive action.  We are not suggesting that these compounds are similarly immunosuppressive, but rather simply have anti-inflammatory action that may have potential as a gentler therapeutic for mild inflammation.
  • The anti-inflammatory effects of compound 3 resulted in a 21.7% decrease in TNF-alpha secretion from macrophages. This is similar to what we have previously found with another anti-inflammatory compound from a Karina   We have included a comparison to this in our discussion.  While a 22% decrease may not appear much, it is important to note that this can translate to a significant physiological response.  The difference in circulating inflammatory cytokines between patients suffering from sepsis and those that are healthy is about 45-50%, and corticosteroids at similar doses have about a 40% reduction.  One could therefore anticipate that a 22% reduction in inflammatory cytokines could have a modest effect that would be physiologically relevant to reduce but not eliminate inflammation.

Reviewer 3 Report

   The article from Leutou and co-workers describes the isolation of two galactolipids and a fatty acid from a dinoflagellate.The evaluation of their antimicrobial and anti-inflammatory activities indicates that the fatty acid exhibits the greatest anti-inflammatory activity. The isolation of natural products with pharmaceutical properties from marine organism is of great importance. Moreover, biochemical analyses of harmful dinoflagellates contributes to a better understanding of the physiology of these organism. Experiments seem to have been well conducted but the introduction and discussion sections of the manuscript could be improved. My comments are as follows:

Title: Include information regarding the evaluation of anti-inflammatory activity

Abstract : The abstract should include information regarding the fatty acid composition of the galactolipids isolated and the PUFAME

Introduction:

MGDG is a plastidial lipid present in photosynthetic organisms, although its fatty acid composition may vary according to the organism. Also C14:0 and C18:5 are fatty acids known to be present in dinoflagellates. Additional information should be provided regarding MGDG molecular species present in dinoflagellates and if available in Karenia. It is not clear why authors consider the MGDG isolated in this work as “new”. Most likely free C18:5 and MGMG re formed by the action of an endogenous acyl hydrolase, which removes fatty acids from MGDG. C18:5 reacts with methanol during the extraction procedures forming the PUFAME.

Line 45: and also that MGDG……..were more potent than those with a myristoyl group at sn-2 position, although…

Line 49: In the abstract the abbreviation MGDG is for monogalactosyldiacylglycerol. The sentence should be rewritten if authors want to compare glucose versus galactose in the head group of glycolipids.

Results and Discussion:

Line 125: results not shown

The results from anti-inflammatory activity should be discussed with more detail by comparing previous experiments with similar compounds from the literature. Since the PUFAME was the compound exhibiting the greatest anti-inflammatory activity authors could explore the possibility of obtaining higher amounts of this compound from the dinoflagellate, since it is probably present also in other polar and neutral lipids.

Experimental section:

“Strain and cultivation” should appear first in this section

Conclusions:

There is no need to define again abbreviations used before in the manuscript (PUFA, MGDG, etc)

Line 286: eliminate “5. Conclusion and line 288

Author Response

Reviewer 3

R3: The article from Leutou and co-workers describes the isolation of two galactolipids and a fatty acid from a dinoflagellate.The evaluation of their antimicrobial and anti-inflammatory activities indicates that the fatty acid exhibits the greatest anti-inflammatory activity. The isolation of natural products with pharmaceutical properties from marine organism is of great importance. Moreover, biochemical analyses of harmful dinoflagellates contributes to a better understanding of the physiology of these organism. Experiments seem to have been well conducted but the introduction and discussion sections of the manuscript could be improved. My comments are as follows:

R3: Title: Include information regarding the evaluation of anti-inflammatory activity

Author’s Response: The title has been changes to reflect the reviewer’s suggestions

R3: Abstract : The abstract should include information regarding the fatty acid composition of the galactolipids isolated and the PUFAME.

Author’s Response: the complete name of the fatty acids and glycolipids have been included in the abstract

Introduction:

R3: MGDG is a plastidial lipid present in photosynthetic organisms, although its fatty acid composition may vary according to the organism. Also C14:0 and C18:5 are fatty acids known to be present in dinoflagellates. Additional information should be provided regarding MGDG molecular species present in dinoflagellates and if available in Karenia. It is not clear why authors consider the MGDG isolated in this work as “new”. Most likely free C18:5 and MGMG re formed by the action of an endogenous acyl hydrolase, which removes fatty acids from MGDG. C18:5 reacts with methanol during the extraction procedures forming the PUFAME.

***Author’s Response: Changes have been made to the Introduction to reflect the reviewers comments. 

R3: Line 45: and also that MGDG……..were more potent than those with a myristoyl group at sn-2 position, although…

Author’s Response: a comma was added after position

R3: Line 49: In the abstract the abbreviation MGDG is for monogalactosyldiacylglycerol. The sentence should be rewritten if authors want to compare glucose versus galactose in the head group of glycolipids.

Author’s response: The compounds that were isolated and described in this paper only had galactose as the head group of the glycolipid.  The sentence regarding different head groups was from a reference, therefore we do not believe that the MGDG acronyms for the compounds described in this paper need to be changed.

Results and Discussion:

R3: Line 125: results not shown

Author’s response: “results not shown” was added to the text.

R3: The results from anti-inflammatory activity should be discussed with more detail by comparing previous experiments with similar compounds from the literature.

 Author’s Response:

  • We appreciate the suggestion of comparing these compounds to an existing anti-inflammatory substance. Unfortunately, there is no good direct comparison of a known anti-inflammatory drug (or positive control) to these types of compounds, because the structure and mechanisms of action of corticosteroids or NSAIDs are very different.  Corticosteriods can reduce cytokine secretion to zero following LPS stimulation, but they also have a myriad of other negative side-effects, including increased susceptibility to infection due to massive immunospressive action.  We are not suggesting that these compounds are similarly immunosuppressive, but rather simply have anti-inflammatory action that may have potential as a gentler therapeutic for mild inflammation.
  • The anti-inflammatory effects of compound 3 resulted in a 21.7% decrease in TNF-alpha secretion from macrophages. This is similar to what we have previously found with another anti-inflammatory compound from a Karina   We have included a comparison to this in our discussion.  While a 22% decrease may not appear much, it is important to note that this can translate to a significant physiological response.  The difference in circulating inflammatory cytokines between patients suffering from sepsis and those that are healthy is about 45-50%, and corticosteroids at similar doses have about a 40% reduction.  One could therefore anticipate that a 22% reduction in inflammatory cytokines could have a modest effect that would be physiologically relevant to reduce but not eliminate inflammation.

Reviewer 3-Since the PUFAME was the compound exhibiting the greatest anti-inflammatory activity authors could explore the possibility of obtaining higher amounts of this compound from the dinoflagellate, since it is probably present also in other polar and neutral lipids.

Author’s response: This is now addressed in the conclusions

Experimental section:

R3: “Strain and cultivation” should appear first in this section

Author’s response: The experimental section was reorganized to flow in a chronological order

Conclusions:

R3: There is no need to define again abbreviations used before in the manuscript (PUFA, MGDG, etc)

Author’s response: The abbreviations were removed from the text

R3: Line 286: eliminate “5. Conclusion and line 288

Author’s response: line 286 the text was eliminated.

Round 2

Reviewer 2 Report

The authors have addressed most of the points raised in my previous review, but there are still some typos. I would recommend this manuscript for publication on this journal after revisions as described below.

Line 76: a known PUFAME, Methyl --> a known PUFAME, methyl

Line 90: 13CNMR --> 13C NMR (a space is required)

Line 124: (MGMG (2) --> (MGMG) (2)

Line 127: HRMSESITOF mass spectrum --> HRESITOF mass spectrum  

Line 139: Monogalactosyldiacylglycerol --> monogalactosyldiacylglycerol

Author Response

Thank you for finding the typos.  All changes have been made

Line 76: a known PUFAME, Methyl --> was changed to  "a known PUFAME, methyl"

Line 90: 13CNMR --> was changed to "13C NMR (a space is required)"

Line 124: (MGMG (2) --> was changed to "(MGMG) (2)"

Line 127: HRMSESITOF mass spectrum --> was changed to "HRESITOF mass spectrum"  

Line 139: Monogalactosyldiacylglycerol --> was changed to "monogalactosyldiacylglycerol"